# Does the Ketogenic Diet Improve the Quality of Ovarian Function in Obese Women?

**DOI:** 10.3390/nu14194147

**Published:** 2022-10-06

**Authors:** Maria Cristina Magagnini, Rosita A. Condorelli, Laura Cimino, Rossella Cannarella, Antonio Aversa, Aldo E. Calogero, Sandro La Vignera

**Affiliations:** 1Department of Clinical and Experimental Medicine, University of Catania, 95123 Catania, Italy; 2Department of Experimental and Clinical Medicine, Magna Graecia University of Catanzaro, 88100 Catanzaro, Italy

**Keywords:** ketogenic diet, ovarian function, obesity, SHBG, AMH

## Abstract

**Background:** Polycystic ovarian syndrome (PCOS) is the most common endocrine disorder in women of reproductive age, the prevalence of which ranges from 8 to 13%. It is characterized by metabolic, reproductive, and psychological alterations. PCOS prevalence is related to body mass index (BMI). Women with BMI < 25 kg/m^2^ have a prevalence of 4.3%, whereas women with BMI > 30 kg/m^2^ have a prevalence of 14%. Moreover, women with PCOS have a risk of type 2 diabetes mellitus (T2DM) two-fold higher than controls, independently of BMI. Both PCOS and T2DM are also consequences of lower serum sex-hormone-binding globulin (SHBG) levels, which is currently considered a biomarker of metabolic disorders, in particular T2DM. **Aim:** To evaluate the effect of the very-low-calorie ketogenic diet (VLCKD) on markers suggested to be predictive of metabolic and ovulatory dysfunction. These comprehend SHBG, anti-Mullerian hormone (AMH), and progesterone levels on day 21 of the menstrual cycle in a cohort of obese non-diabetic women with PCOS and regular menses. **Methods:** Twenty-five patients (mean age 25.4 ± 3.44 years) with obesity and PCOS underwent VLCKD for 12 weeks. Each of them underwent measurements of anthropometric parameters (body weight, height, and waist circumference) and blood testing to evaluate serum levels of SHBG, AMH, and progesterone before and after 12 weeks of VLCKD. **Results:** At enrollment, all patients had high BMI, WC, and AMH, whereas SHBG and progesterone levels were low. After VLCKD, the patients showed a significant reduction in BMI, WC, and HOMA index. In particular, 76% of patients (19/25) switched from obesity to overweight, and the HOMA index normalized, reaching values lower than 2.5 in 96% (24/25) of patients. In addition, serum AMH levels significantly decreased, and progesterone and SHBG significantly increased after VLCKD. **Conclusions:** This is the first study documenting the effects of VLCKD on ovarian reserve and luteal function in women with PCOS. VLCKD could be used to improve metabolic and ovulatory dysfunction in women with PCOS. Further studies are needed to understand the reasons for the AMH reduction.

## 1. Introduction

Weight loss could be the key to improving the metabolic profile and reducing the endocrine dysfunction responsible for infertility. Nutrition, in fact, could play an important role in improving reproductive outcomes [1].

Several studies have demonstrated the benefits of the Mediterranean diet (MetD) (rich in fiber, omega-3 fatty acids, vitamins and minerals) on the female reproductive system. In particular, it seems that MetD is associated with a reduction in the insulin resistance index and an improvement in metabolic parameters [2], an increase in pregnancy and live birth, in particular in women <35 years old, and an increased probability of live birth in women who undergo assisted reproductive technique (ART) [3].

Missmer and colleagues, in a cohort study including 17,544 women planning pregnancy, found that there is a correlation between a pro-fertility diet (similar to the MetD and characterized by a high consumption of fiber, low-glycemic-index foods, dairy products, vegetable proteins, and monounsaturated fatty acids and a reduced consumption of animal proteins and trans fatty acids) and a low risk of infertility due to ovulatory disorders [4]. Moreover, these women, took multivitamins (e.g., acid folic) at least three times/week and exercised regularly [2,3].

In contrast to the MetD, there is the Western-style diet, characterized by high consumption of simple sugars and products with a high glycemic index, saturated fats, red meat, and a low intake of fresh fruit, vegetables, fish, fiber, and unrefined grains. Several studies have shown that a diet based on a high content of sugars, saturated fats, and animal proteins negatively impacts fertility and in particular is responsible for alterations in the menstrual cycle, with reduced production of progesterone and anti-Müllerian hormone (AMH), a greater number of antral follicles, and fewer blastocysts [2].

As explained above, insulin resistance is one of the factors that aggravate oocyte quality and maturation and damages female fertility. The food intake with a high glycemic index is associated with a higher concentration of fasting glucose, androgens, and insulin growth factor-1 (IGF1) and can worsen insulin resistance and, therefore, ovulatory outcomes; not only that, it can be the cause of increased glucotoxicity and, therefore, of increased oxidative stress, which further damages ovarian function [2,5]. Several studies have shown that a diet based on high-glycemic-index carbohydrates is associated with ovulatory disorders and a reduced chance of becoming pregnant [2]. For example, in women with polycystic ovarian syndrome (PCOS), a low carbohydrate diet could represent an alternative to pharmacological treatment with metformin, as it can reduce the stimulus on the release of insulin of the pancreatic cell by glucose. This was also confirmed in normal-weight PCOS women [6].

As far as fatty acids are concerned, a diet with a high lipid content can be associated with an alteration in the menstrual cycle and hormonal values, as well as an altered oocyte quality [2]. The definition of a high-fat diet (HFD) varies from 30% to 75% of total daily lipid intake. When the intake of lipids is excessive, they are also deposited in non-adipose tissues and create a state of lipotoxicity, which damages the reproductive system [7]. It has been seen that in the case of HFD there is: a transient but significant increase in LH levels; a reduction in the pool of primordial follicles and an increase in follicular atresia, with an increased risk of premature ovarian insufficiency (POI); an increase in oocyte lipotoxicity with a higher rate of anovulation; and an alteration of ovarian steroidogenesis by downregulation of aromatase [7].

Regarding the high trans fatty acids (TFA) content, there are conflicting studies. Chavarro and colleagues, in the previously described study, conducted in the USA from 1991 to 1995, showed that even a 2% increase in TFAs is responsible for an increased rate of infertility for ovulatory disorders [8]. However, Mumford and colleagues, in the Byocicle study, conducted in the USA from 2005 to 2007, found no association between the amount of TFAs and anovulatory disorders [9]. Probably this discrepancy is due to the fact that, in 2005, foods in the United States had to be labeled when the TFA content was >0.5 g. TFAs have a proinflammatory effect, increase the insulin resistance index, and reduce the expression of the Peroxisome proliferator- activated receptor gamma (PPAR-γ) receptor. PPAR-γ is a member of the nuclear receptor superfamily. It is involved in many metabolic processes, such as lipid metabolism and insulin sensitivity. HFD alters PPAR-γ conformation and its gene expression, reducing insulin sensitivity [7].

Conversely, Mono Unsaturated Fatty Acids (MUFAs) bind to the PPAR-γ reducing inflammation, increasing insulin sensitivity, improving fertility, and reducing the time to pregnancy. The World Health Organization has recommended a reduction in the consumption of foods with TFAs, which must represent less than 1% of the total daily intake. Moreover, TFAs increase risk of type 2 diabetes, PCOS, and endometriosis [2].

A defensive role in fertility seems to be played by omega 3 fat acids (FAs), which we find mainly in fish oil. They have an anti-inflammatory effect and act directly on steroidogenesis, increasing the concentration of progesterone; promoting the maturation of the oocyte and improving the quality of the embryo; reducing insulin resistance; and improving the lipid profile. Furthermore, it has been shown that a higher omega 3 content is associated with a higher rate of live births after ART [2].

Finally, again, the study by Chavarro and colleagues shows how a low consumption of dairy products (milk, cheese, and yogurt) is responsible for a reduced fertility rate. This is probably due to the fact that dairy products contain a high content of estrogen, fat-soluble vitamins, and trans-palmitoleic acid, which improves insulin sensitivity [4]. However, there are also studies in which the direct correlation between increased consumption of dairy products and increased fertility is not confirmed. Taking more than three servings of dairy products per week reduces the risk of endometriosis by 18%. Finally, increased consumption of dairy per day is associated with an increase in live births in women over 35 years of age. In conclusion, in women wishing to become pregnant, the intake of fatty acids must be based on a preference for Poly Unsaturated Fatty Acids (PUFAs) and MUFAs and a limited intake of TFAs and saturated FAs (SFAs) [2].

The role that proteins play within reproductive mechanisms is not yet clear. Mumford and colleagues showed that excess animal protein intake correlates with decreased testosterone levels [9]. The Chavarro study found that protein consumption also affects fertility. In particular, it has shown that animal proteins negatively impact ovulation as they facilitate the release of insulin and IGF1 [2,8]. This is not the case with plant proteins, which increase the fertility rate in women >32 years, which has not yet been identified [6].

Accordingly, infertile obese women have worse outcomes in the case of assisted reproductive technique (ART), as they need a higher dose of gonadotropins, have a greater risk of ovarian hyperstimulation, increased cycle cancellation rate, and lower oocyte recovery, and increased abortion rate. Live birth rate is the most significant parameter when using ART. A systematic review and meta-analysis conducted by Sermondade and colleagues in 2019 evaluated the association between female obesity and the probability of live birth rate following ART [10]. This meta-analysis, which evaluated 682,532 cycles, showed that female obesity negatively impacts the live birth rate following ART [10].

Several studies have demonstrated the benefits of the Mediterranean diet (MetD) (rich in fiber, omega-3 fatty acids, vitamins and minerals) on the female reproductive system. In particular, it seems that MetD is associated with a reduction in the insulin resistance index and an improvement in metabolic parameters [2]; an increase in pregnancy and live birth, in particular in women <35 years old; and an increased probability of live birth in women who undergo ART [3]. A prospective cohort study conducted by Hongmei and colleagues evaluated the effects of the MedD on ART outcomes. The women involved were divided into two groups: the higher MetD adherence group and lower MetD adherence group. No significant differences were found in baseline characteristics between groups. Instead, the group with the highest adherence to the MetD showed a positive correlation with the yield and availability of fertilized oocytes and embryos, while not increasing the success rate of ART. These results contrast with those described by Karayiannis and colleagues in 2018 and by Vujkovic and colleagues in 2010 who, through their studies, had seen how greater adherence to the MetD is associated with a higher pregnancy rate in ART cycles but not with an improvement in fertilization rate and embryo yield [10]. Finally, Grieger and colleagues have shown that a diet rich in omega-3 can affect the quality of the embryo following ART [6]. However, weight loss in overweight or obese women who will undergo ART improves fertility and the outcome of pregnancy. In particular, there is greater menstrual regularity, qualitative improvement of the embryo to be transferred, lower dose of gonadotropins needed, and reduced number of treatment cycles [11].

From the studies conducted to date, it seems that a diet with low glycemic index foods, rich in mono and polyunsaturated fats, and which favors vegetable proteins is the most suitable for obese and overweight women who wish to become pregnant, both to improve endocrine dysfunctions of the female reproductive system and to ensure a better outcome of the maternal–fetal aspects of pregnancy.

However, there are no specific guidelines regarding the type of diet to be carried out to lose weight in obese women who wish to become pregnant [12].

Today, a particular nutritional protocol that is becoming increasingly popular to promote weight loss and improve fertility outcomes is the ketogenic diet (KD). KD is a nutritional protocol that mimics fasting through a marked restriction of daily carbohydrate intake (<30 g/per day, that is, 13% of total energy) and a proportional intake of fat (about 44%) and protein (about 43%). It is not a high-protein diet, in fact, daily protein intake is 1.2–1.5 g/kg of ideal body weight [13]. One of the advantages of the ketosis state is the capability of ketone bodies of reducing the appetite, due to the inhibition of the release of cerebral neuropeptide Y and ghrelin.

Ketone bodies (3β-hydroxybutyrate, acetoacetate, and acetone) are produced in hepatocyte mitochondria in some specific condition, in which there is reduced availability of glucose, such as fasting, intensive physical activity, and very-low-calorie ketogenic diet (VLCKD) [13]. Ketone bodies, in fact, represent a source of alternative energy for organs (heart, cerebral tissue, skeletal muscle, and kidney) [14,15]. In physiological conditions, oxaloacetate, derived from pyruvate during glycolysis, enters in the tricarboxylic acid cycle (TCA), condensing with acetyl-CoA and forming citrate. If the levels of glucose are low or glycolysis is stopped (fasting), the acetyl-CoA’s excess, derived from the oxidation of fat acids, cannot enter in TCA but is transported to the hepatocyte mitochondria and transformed in ketone bodies, by condensing of two acetyl-CoA molecules into acetoacetyl-CoA [13]. Then, ketone bodies are transported in peripheral organs to be oxidized and to produce energy [14,15]. In conditions of reduced availability of glucose, the liver uses the fatty acids mobilized from the adipose deposits and the ketogenic amino acids to produce ketone bodies: acetoacetate, 3-β-hydroxybutyrate, and acetone.

Simulating fasting (even in the absence of caloric restriction) has specific effects on numerous energy-dependent pathways, including the inhibition of the pathways mediated by IGF-1 and mTOR. Furthermore, 3-β-hydroxybutyrate, in addition to being an energy substrate, also represents a molecule with signaling and epigenetic functions, with modulatory effects on inflammation and oxidative stress. Moreover, KD promotes changes in the metabolic structure and in some molecular processes with numerous advantages: increasing insulin sensitivity, inhibiting hydroxymethylglutaryl-CoA reductase, reducing lipid synthesis, reducing inflammatory pathways, protecting from oxidative stress, and inducing autophagy and mitochondrion genesis. Therefore, it finds application in some conditions of female hormonal imbalance in the presence of ovulatory disorders or overweight/obesity. In fact, it appears to be particularly effective in overweight and obese women with PCOS, in which it was highlighted that after KD there is a 12% reduction in total weight, a 54% reduction in insulin levels, a 36% reversal of the luteinizing hormone (LH)/follicle-stimulating hormone (FSH) ratio, and a 22% reduction in free testosterone. In addition, about 24 weeks after the study ended, 40% of patients who had completed the KD protocol became pregnant [16]. One of the most important effects of KD is on mitochondrial function, which is responsible for poor oocyte quality in obese women. Mitochondria play a vital role in oocyte quality. Its compromise is that it is related to infertility, to a lower number of cells in the blastocyst, and to the loss of the embryo. Furthermore, oxidative stress and mitochondrial dysfunctions in women with PCOS are responsible for: reduced synthesis of sex hormone binding globulin (SHBG) synthesis, chronic inflammatory state, poor oocyte quality, increased androgen production, and reduced progesterone production. Several studies have shown that KD is able to restore mitochondrial function, and it does so independently of weight loss [16].

The aim of this study was to evaluate the 3-month-long impact of VLCKD on AMH levels (ovarian reserve quality marker), progesterone levels on the 21st day of menstrual cycle (luteal phase adequacy marker), and SHBG levels (prognostic metabolic risk marker for type 2 diabetes mellitus (T2DM) development) in a cohort of obese non-diabetic women with PCOS and regular menstruation. To accomplish this, we retrospectively reviewed the clinical charts of the patients attending the Endocrine Division, University-Teaching hospital Policlinico “G. Rodolico—San Marco”, for obesity and infertility.

## 2. Patients and Methods

### 2.1. Patient Selection

This is a retrospective study performed on women who consulted the Division of Endocrinology, Metabolic Disease and Nutrition, University of Catania, for body weight increase. All PCOS patients that had first-degree obesity (BMI 30–34.9 kg/m^2^) and underwent VLCKD, from January 2021 to January 2022, were consecutively enrolled in this study. We consecutively recruited twenty-five Caucasian women older than 18 years (mean age 25.4 ± 3.4 years). PCOS was diagnosed using at least two criteria of the 2003 Rotterdam Consensus (hyperandrogenism, ultrasonographic view of polycystic ovaries and oligo/anovulation) [17], family history of T2DM (mother or father), regular menstrual cycle intervals, and male partners with normozoospermia. The patients enrolled did not have: type 1 diabetes mellitus (T1DM), latent autoimmune diabetes, T2DM, chronic renal failure (namely, those with estimated glomerular filtration rate <60 mL/min/1.73 mq), active or severe infections, recent major cardiovascular event, unstable angina, cardiac arrhythmias, frailty, 48 h prior surgery or invasive procedures, eating disorders and other psychiatric disturbances, partner with oligoasthenoteratozoospermia, or irregularity of menstrual cycles.

### 2.2. Hormonal Measurement

Anthropometric and biochemical parameters were evaluated at enrollment and at the end of the second phase (after 12 weeks from the beginning of VLCKD). ß-hydroxybutyrate levels were measured weekly through a reflex metric detection system and kept between 0.5 and 0.7 mmol/L. Body weight was measured in fasting conditions, using the same calibrated scale, without shoes, and with an empty bladder.

Each woman underwent blood testing for the measurement of AMH, SHBG, and progesterone levels on the 21st day of the menstrual cycle, before and after 12 weeks of VLCKD. The hormones were analyzed immediately, and the samples were not frozen.

The hormone evaluation was performed by Cobas Elecsys for AMH (measurement range: 0.000001–23 ng/mL) and SHBG (measurement range: 32.4–128 nmol/L) and by Cobas Elecsys III for Progesterone (measurement range: 1.83–23.9 ng/mL).

### 2.3. VLCKD Protocol

The VLCKD protocol uses high-biological protein derived from milk, peas, and whey. Each preparation, which has about 100–150 kcal, contains 18 g of protein, 4 g of carbohydrate, and 3 or 4 g of fat. It is based on 3 stages: active stage, which represents the state of VLCKD (600–800 kcal/day) and includes the first three phases; re-education stage, which represents the state of LC diet (1200–1500 kcal/day) and includes from phases 4 to phases 6; and maintenance stage, that is, the balanced diet (1500–2000 kcal/day) and includes the seventh and last phase. Each phase had a duration of 4 weeks.

In the first phase, there is a total replacement of natural protein with protein preparation (4 meals for the woman and 5 meals for the man), with the possibility of consuming low-glycemic-index vegetables during lunch and dinner.

In the second phase, one protein preparation is replaced by natural protein food (lunch or dinner), such as meat, egg, or fish (always with a low glycemic index vegetables). In the third phase, both lunch and dinner protein preparations are replaced with natural proteins. During the active stage, it is recommended to integrate with micronutrients, such as vitamins (complex B, C, and E), minerals (sodium, potassium, magnesium, and calcium) and omega-3 fatty acid. The re-education stage is a low-calorie diet during which carbohydrates are progressively re-integrated according to their increasing glycemic index: fruit and dairy products in the fourth phase; legumes in the fifth phase; and bread, pasta and cereal in the last phase, with a daily intake between 800 and 1500 kcal. Finally, during the maintenance stage, there is an eating plan balanced in macro- and micronutrients with a daily intake of between 1500 and 2000 kcal/day, depending on the individual [14].

During VLCKD, patients must be closely and periodically monitored [13] (Table 1) to prevent complications, such as dehydration or hydro-electrolytic imbalances.

### 2.4. Statistical Analysis

Continuous variables are reported as mean ± SD or median (interquartile range) according to their distribution. Normality was evaluated using the Shapiro–Wilk test. Analysis of data was hence performed by the Student’s 𝑡-test or the Wilcoxon test, according to their distribution.

Statistical analysis was performed using MedCalc Software Ltd. (Version 19.6–64 bit) Ostend, Belgium. A *p*-value of less than 0.05 was accepted as statistically significant. A trend was assumed for *p*-values ranging from 0.05 to 0.099.

## 3. Results

All patients enrolled in the study completed the VLCKD protocol, and no drop-out was registered. Baseline parameters are detailed in Table 2.

At enrollment, the mean waist circumference (WC) and body mass index (BMI) were 88.4 ± 3.2 cm and 32.8 ± 1.0 Kg/m^2^, respectively. The values of HOMA index were 5.6 ± 0.8, indicating the presence of insulin resistance. Furthermore, AMH and progesterone levels on the 21st day of the menstrual cycle were 7.8 ± 3.0 ng/mL and 3.4 ± 1.1 ng/mL, respectively. The pre-VLCKD SHBG serum levels were 5.9 ± 4.6 nmol/L.

After the 3-month-long VLCKD, the patients showed a significant reduction in the WC (Figure 1A), BMI (Figure 1B), and HOMA index (Figure 1C). In particular, 76% of patients (19/25) were no longer obese but became overweight, and the HOMA index normalized, reaching values lower than 2.5 in 96% (24/25) of patients. The mean weight loss achieved was 18 ± 4.2 kg. Furthermore, AMH serum levels significantly decreased (Figure 1D), whereas progesterone (Figure 1E) and SHBG (Figure 1F) significantly increased after VLCKD. Finally, the percentage of patients with progesterone levels at the 21st day of menstrual cycle > 15.9 ng/mL (considered by Chinta and colleagues as a sign of ovulation [18]) was 100%.

## 4. Discussion

The present study was conducted to evaluate the effects of VLCKD on metabolic and ovulatory disorders in a cohort of obese non-diabetic women with PCOS and regular menstruation, family history of T2DM (mother or father), and male partners with normozoospermia. The diagnosis of PCOS was made using the criteria established by the 2003 Rotterdam Consensus (oligomenorrhea/anovulation hyperandrogenism, and ultrasound appearance of polycystic ovaries) [17].

PCOS is the most common endocrine disorder in women of reproductive age and the most frequent cause of anovulatory infertility. It is characterized by metabolic, reproductive, and psychological alterations. The prevalence of PCOS is correlated with BMI. Women with BMI < 25 kg/m^2^ have a prevalence of 4.3%, whereas women with BMI > 30 kg/m^2^ have a prevalence of 14%. Moreover, PCOS women have a risk of T2DM two-fold-higher than controls, independently of the BMI. This condition is also exacerbated by the reduction in SHBG, which is currently considered a biomarker of metabolic disorders, in particular T2DM. To date, in fact, several studies have demonstrated how SHBG also represents a biomarker of metabolic disorders, such as de novo lipogenesis. This is the formation of fatty acid from non-lipid precursors, such as glucose, and is one of the principal pathways involved in the accumulation of intrahepatic lipids. SHBG, by downregulating de novo lipogenesis, reduces intrahepatic lipids, which increase in the presence of T2DM, obesity, and PCOS [19]. Conversely, serum SHBG levels are reduced in T2DM, obesity, and PCOS.

Our study highlights that 3-month-long VLCKD had a significant reduction in the BMI and a significant improvement in the SHBG levels, therefore supporting the presence of an improvement in metabolic parameters, which is also evident with the reduction in BMI, waist circumference, and the HOMA index.

For a long time, it was thought that the common denominator of reduced SHBG synthesis was hyperinsulinemia. More recent studies, however, have shown that the transmission of SHBG-altering polymorphisms (single nucleotide polymorphisms in the *SHBG* gene) is associated with the risk of developing T2DM. These findings suggest that SHBG probably has a direct physiological role in glucose homeostasis [20] and strengthens the hypothesis that reduction in SHBG may predict future risk of T2DM in both women and men [21]. Furthermore, a study showed that genetically determined SHBG was also inversely associated with PCOS [19]. In fact, PCOS women have lower serum levels of SHBG compared with women without PCOS, and this contributes to a greater bioavailability of androgens and a worsening of the symptoms associated with them. Moreover, women with PCOS have insulin resistance that is thought to be intrinsic to the syndrome, which contributes to a worsening of the metabolic pathophysiology of PCOS and reduces SHBG levels. For these reasons, they have an increased risk to develop T2DM [20].

The association between low SHBG levels and increased risk of T2DM could be due to the fact that SHBG appears to be a biomarker for high insulin and blood glucose levels [22]. Numerous cross-sectional studies have shown a relationship between serum SHBG levels and T2DM. These studies found that diabetic women had significantly lower levels of SHBG compared with non-diabetic women. Moreover, SHBG levels were inversely correlated with glycated hemoglobin, suggesting that SHBG levels seem to be associated with alteration in glucose homeostasis, even before the development of diabetes. In particular, Ding and colleagues have reported that women with SHBG values above 60 nmol/L have an 80% reduction in the likelihood of developing T2DM [22], giving further confirmation of the correlation between SHBG, glycemic homeostasis, and risk of T2DM [22]. Finally, many studies have shown a direct correlation between sex hormone levels and T2DM. Ding and colleagues demonstrated a sexually dimorphic relationship between testosterone levels and the risk of developing T2DM: low testosterone levels are associated with a high risk in men, while high levels are associated with a high risk in women, probably because the synthesis of SHBG depends on numerous factors, both metabolic and hormonal, including insulin and sex steroids.

Given the hyperandrogenism and insulin resistance condition of PCOS, it has been proposed that mutations of the genes responsible for SHBG synthesis may also be responsible for PCOS. In this regard, numerous studies have been conducted, but none have been able to prove this specific correlation to date, and the studies are also inconclusive [22]. Ruth and colleagues also conducted a Mendelian randomization study to evaluate the relationship between serum SHBG, total testosterone, and free testosterone and the presence of diseases, such as T2DM and PCOS [23]. In women, this Mendelian randomization study suggested a causal association of SHBG and free testosterone with T2DM, but not with total testosterone. Several studies have shown that the lifestyle and in particular the MetD and its components bring direct benefits to the levels of SHBG (as hepatokine), causing an increase in serum levels. In particular, a 10 kg weight loss results in a 26% increase in SHBG. However, 20% of this increase depends on the reduction in the intrahepatic lipid content. A target diet for de novo lipogenesis is the low-carbohydrate diet—such as VLCKD, which appears to be more effective than the classic MetD. Lifestyle changes not only have a direct effect on SHBG as a hepatokine but also a metabolic effect on the de novo lipogenesis pathway (intrahepatic accumulation of lipids and insulin resistance), of which SHBG is a biomarker [19].

Our study shows that after three months of VLCKD and a significant weight loss, the AMH value is significantly reduced, normalizing by age. AMH serum levels represent a marker of the ovarian reserve. Until 2018, the ultrasound criteria for making the diagnosis followed the Rotterdam criteria, in which at least one ovary with 12 follicles of 2–9 mm or a volume > 10 mL in the absence of a dominant follicle (diameter > 10 mm) [24]. Today, however, as technology has improved, the ability to see the follicles has improved, so the cutoffs changed. The last guidelines recommended using follicle number per ovary (FNPO ≥ 20 follicles per ovary) for the definition of polycystic ovary morphology (PCOM) and ≥25 if using a high-resolution ultrasound system. While only if advanced ultrasound technology is not available, the guidelines recommended ovarian volume ≥10 mL for diagnosis of PCOM [25]. A potential alternative factor to ultrasound PCOM is AMH, whose serum levels are significantly higher in women with PCOS (ranging from 20 to 81.6 pmol/L) compared with women without PCOS and regular ovulation (ranging from 16.7 to 33.5 pmol/L) [25,26]. To date, the hypothesis is coming forward that AMH can be considered valid and independent marker or even an alternative to follicular count for the diagnosis of PCOS. Unfortunately, the measurement of AMH has numerous technical limitations. Furthermore, its values vary both with age and in the presence of diseases, such as hypogonadotropic hypogonadism. It is for this reason that its measurement cannot be taken into consideration as an isolated diagnostic parameter but can only be used to support the suspected presence of PCOS [25].

The reduction in the AMH levels found in the present study may possibly indicate an improvement in the ovulation rate, due to both the increase in SHBG (with reduction in bioavailable androgens) and the reduction in insulin levels (and consequent improvement in LH pulsatility and further reduction in androgens). To date, however, there are no studies that demonstrate that weight loss has beneficial effects on serum levels of AMH, but above all, it is not certain that normalization can be a sign of better ovulatory outcome or a lower rate of ovarian hyperstimulation syndrome in the course of ART. In fact, AMH is a negative prognostic factor during ovarian stimulation cycles: the higher it is, the greater the probability of incurring in a hyperstimulation syndrome (AMH > 3.4 ng/mL).

It is known that an adequate production of progesterone in the luteal phase is necessary to improve the secretory phase of the endometrium, which plays a relevant role for the implantation of the embryo. In our study, the patients enrolled had a regular cycle but low levels of progesterone in the mid-luteal phase. After 3 months of VLCKD, we achieved a significant increase in progesterone levels on day 21 of the menstrual cycle, obtaining an improvement in ovulatory dysfunction.

To date, there is no specific definition of luteal phase insufficiency. Previous investigations have suggested cutoffs ranging from 10 to 15 ng/mL to differentiate low from normal mid-luteal progesterone levels [18]. The AMIGOS trial, a prospective, multicenter, randomized clinical trial that evaluated conception, clinical pregnancy, live birth, and multiple gestation rates associated with ovarian stimulation-IUI in couples with unexplained infertility, demonstrated an increase in the live birth rate in all treatment groups with higher mid-luteal progesterone levels; meanwhile, there were no live births in the clomiphene citrate cycles when the mid-luteal progesterone level was 14.4 ng/mL, in the letrozole cycles when it was 13.1 ng/mL, and in the gonadotropin cycles when it was 4.3 ng/mL. In addition, there was a lower probability of obtaining a live birth in the clomiphene citrate cycles when the mid-luteal progesterone level was <15.8 ng/mL, in the letrozole cycles when it was 14.8 ng/mL, and in the gonadotropin cycles when it was 9.1 ng/mL.

This is probably due to the fact that higher progesterone levels are necessary to induce appropriate endometrial receptivity. The negative predictive value of a low mid-luteal progesterone cycle was 94.9% [18].

Lower progesterone levels were more common in women with higher BMI, and they were also associated with longer follicular phases than in normal progesterone levels, as if they reflected a lower follicular count or an altered granulosa cell function. Moreover, it has been shown that progesterone also intervenes in the acrosomal reaction of the spermatozoa, playing a crucial role in the intracellular signaling cascade. The acrosome reaction is a calcium-dependent reaction, fundamental in fertilization [22]. It allows spermatozoa to penetrate the zona pellucida and fuse with the oocyte membrane [22,27]. Therefore, a progesterone deficiency in the mid-luteal phase could be the cause not only of a reduced endometrial receptivity but also of an altered acrosome reaction.

Women with PCOS and regular cycles can undergo anovulatory cycles. In these cases, the measurement of mid-luteal serum progesterone may help as an additional screening test [24]. In fact, The U.K. National Institute for Health and Clinical Excellence (NICE) guidelines recommend mid-luteal progesterone testing for confirming ovulation even in women with regular menstruation [18].

This is the first study that correlates weight loss with a qualitative and quantitative improvement in a mid-luteal phase marker. Not having a control group with another nutritional protocol, it cannot be said that the increase in progesterone is exclusively due to VLCKD, but it can certainly be deduced that weight loss can improve not only metabolic parameters but also ovulatory ones.

## 5. Conclusions

This is the first study documenting the effects of VLCKD on ovarian reserve and luteal function in obese women with PCOS. VLCKD can be considered a safe dietary approach and can be included among the therapeutic strategies for the metabolic and ovulatory improvement of women with PCOS. Despite this, having no control group, we cannot be sure that the benefits obtained are due to the ketogenic diet or weight loss itself. Further controlled studies on a larger sample size are needed to confirm our findings. Certainly, one of the advantages of VLCKD is that the metabolic and ovulatory improvement is achieved in a relatively short time, and this should be taken into consideration in obese women who have to undergo ART.

## Figures and Tables

**Figure 1 nutrients-14-04147-f001:**
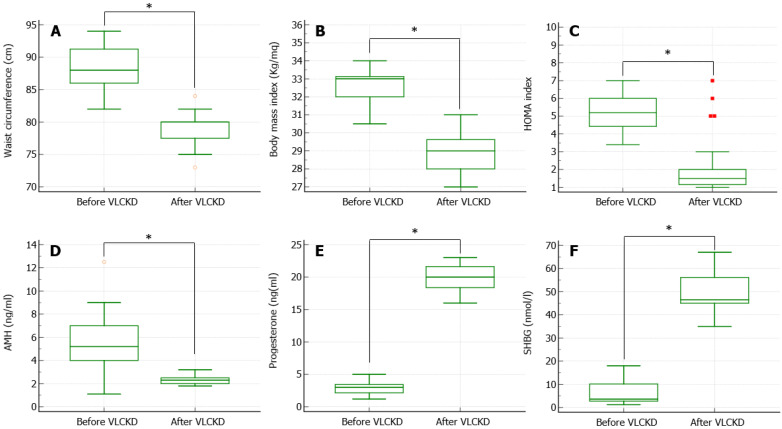
Anthropometric, metabolic, and hormonal parameters measured before and after a 3-month-long period of very low calorie ketogenic diet (VLCKD). (**A**) shows a significant reduction in the waist circumference. (**B**) shows a significant reduction in body mass index. (**C**) shows a significant reduction in insulin resistance, measured using the Homeostatic Model Assessment (HOMA) for Insulin Resistance index. (**D**) shows a significant reduction in the anti-Müllerian hormone (AMH) serum levels. (**E**) shows a significant increase in the progesterone serum levels, measured at the 21st day of the menstrual cycle. (**F**) shows a significant increase in the sex-hormone binding globulin (SHBG) serum levels. * *p* < 0.05.

**Table 1 nutrients-14-04147-t001:** Variable that must be monitored in patients undergoing VLCKD.

Parameter	Baseline	During Active Stages	At the and of VLCKD
Weight, height, body mass index.	Yes	Yes	Yes
Body composition and hydration status (by bioelectrical impedance analysis).	Yes	Yes	Yes
Complete blood count with platelets. Sodium, potassium, magnesium, inorganic phosphate. Albumin, aspartate aminotransferase, alanine transaminase, blood urea nitrogen, creatinine, glutamiltranferase, total bilirubin. Fasting lipid profile ^. 25-hydroxy vitamin D ^, calcium ^. Glucose ^, insulin ^. B-Hydroxybutyrate *. Thyroid stimulating hormone §, free thyroxine §. Complete urinalysis.	Yes	Yes	Yes

* not at baseline and at the end of VLCKD; ^ not during active stages; § not during active stages and at the end of VLCKD.

**Table 2 nutrients-14-04147-t002:** Baseline parameters of the enrolled cohort.

Parameter	Baseline	Percentage of Abnormal Parameters at Baseline
Age (years)	25.4 ± 3.4	-
WC (cm)	88.4 ± 3.2	>88 cm: 48% (12/25)
Weight (Kg)	93 ± 6.2	>80 kg: 100% (25/25)
BMI (Kg/mq)	32.8 ± 1.0	>30 Kg/mq: 100% (25/25)
HOMA index	5.6 ± 0.8	>2.5: 100% (25/25)
AMH (ng/mL)	7.8 ± 3.0	2.5 ± 0.9
Progesterone (ng/mL) *	3.4 ± 1.1	>15.9 ng/mL: 0% (0/25)
SHBG (nmol/L)	5.9 ± 4.6	<32.4 nmol/L: 100% (25/25)

* measured on the 21st day of the menstrual cycle.

## Data Availability

Data are available upon request to the corresponding author.

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
