# Peer review of "Does the Ketogenic Diet Improve the Quality of Ovarian Function in Obese Women?"

_nutrients, 2022, doi:10.3390/nu14194147_

Round 1

Reviewer 1 Report

Only limited data on weight loss and fertility. 

More of an audit of their practice than a formal study. 

The introduction and discussion are far too long and read like a review rather than an original article introducing and discussing a specific study. Both sections should be dramatically reduced and focused on what is needed to introduce and discuss the study at hand. 

The results are not presented in their entirety eg baseline characteristics, weight loss achieved etc and further tables should be added.

Specific points: 

Line 13: Women with PCOS

Line 96 intake rather than income

Line 120 : ? why has not been identified

Line 136: ? and available

Line 142: weight loss in women (repeated)

Line 145: [lower] dose of gonadotrophins needed

Line 160: orexizing?

Line 173: peripheral organs to be oxidate [oxidized}

Line 178: via mediated ?

Lin 157 per day rather than die. 

Line 157 44%+43% adds up to 87%. Is the carb % as intended? Seems high. 

Line 163 specific rather than specifical

Line 165 noble?

Line 160: orexizing? oxidate?

Line 173: peripheral organs to be oxidate [oxidized}

Line 177 caloric restriction 

Line 178: via mediated ?

Line 178 inhibition of… 

Line 205 – states regular menstrual cycles, but line 217 states by Rotterdam. Were women with regular menstrual cycles included or not?? Was it based on anovulation ie a progesterone level < … 

All baseline characteristics of the cohort should be presented in a baseline characteristics table eg AMH, cycle length, androgens etc

Line 218 why would family history of T2DM be an essential criteria for the cohort ?

Line 224 : incomplete

Line 237 just to clarify the bloods were analysed immediately rather than being frozen. 

Line 248 please clarify the duration of each phase. 

Line 263 please present the results of this study. 

Figure 1- Can you also present % with progesterone >15.9 nmol/L as per Leiva reference on threshold for ovulation. Please state how outliers were defined. 

Why are Mean ±SD used in the text, but median plotted on box plots. What was the distribution of the data parametric ? 

Line 301 oligomerrhea spelt wrong and did patients not have regular cycles in this study?

Line 309 did you intend to say 'exacerbated' by rather being a 'consequence' as such. Low SHBG is a consequence of PCOS and obesity / insulin resistance also, but in turn this increase free androgens. I see you are using a mendelian randomisation study to suggest causality, but there remains a bidirectional relationship which is well recognised.

Line 310: which is currently, considered 

Lin 362 how much weight was lost in this study? Please add to results table. 

Line 392 and 394 Add refs

Author Response

Answers to Reviewer #1 comments

Manuscript ID: nutrients-1867796

Comment 1: Only limited data on weight loss and fertility. More of an audit of their practice than a formal study. 

Answer to comment 1: Dear Reviewer, first of all we would like to thank you for the time you spent in reviewing our article, as well as for your constructive criticism.

Secondly, single-center experiences are often reported in literature (https://pubmed.ncbi.nlm.nih.gov/?term=a+single+center+experience&sort=date) and can represent a good source of data in fields where the need of evidence is urgent. VLCKD is being used in the last few years, apparently with exceptional results. Understanding its effects on several diseases is important to provide clinical guidance. Since the evidence on the effects of VLCKD in women with PCOS is lacking, we strongly believe that the data coming from our experience can be useful for the clinical and scientific community, and can inspire other scientists to carry out further (hopefully) multicenter studies on this.   

Comment 2: The introduction and discussion are far too long and read like a review rather than an original article introducing and discussing a specific study. Both sections should be dramatically reduced and focused on what is needed to introduce and discuss the study at hand. 

Answer to comment 2: We agree with you. These sections have been reduced accordingly. Thank you.

Comment 3: The results are not presented in their entirety eg baseline characteristics, weight loss achieved etc and further tables should be added.

Answer to comment 3: Baseline characteristics have been tabulated in table 2 of the revised article. We have also included the mean weight loss (please see lines 299-300).

Comment 4: Specific points: Line 13: Women with PCOS

Answer to comment 4: Modified accordingly.

Comment 5: Line 96 intake rather than income

Answer to comment 5: Changed as requested.

Comment 6: Line 120 : ? why has not been identified

Answer to comment 6:

Comment 7: Line 136: ? and available

Answer to comment 7: We rephrased the sentence to make it clearer.

Comment 8: Line 142: weight loss in women (repeated)

Answer to comment 8: Corrected.

Comment 9: Line 145: [lower] dose of gonadotrophins needed

Answer to comment 9: Added.

Comment 10: Line 160: orexizing?

Answer to comment 10: The sentence has been rephrased as follows: One of the advantages of the ketosis state is the capability of ketone bodies of reducing the appetite, due to the inhibition of the release of cerebral neuropeptide Y and ghrelin.

Comment 11: Line 173: peripheral organs to be oxidate [oxidized}

Answer to comment 11: Corrected accordingly.

Comment 12: Line 178: via mediated ?

Answer to comment 12: Corrected into: “…inhibition of the pathways mediated by IGF-1 and mTOR”

Comment 13: Lin 157 per day rather than die. 

Answer to comment 13: Modified, as requested.

Comment 14: Line 157 44%+43% adds up to 87%. Is the carb % as intended? Seems high. 

Answer to comment 14: Thank you for pointing out this. It is 13% from carb, 44% from fat, 43% from protein. This has been corrected accordingly.

Comment 15: Line 163 specific rather than specifical

Answer to comment 15: Corrected.

Comment 16: Line 165 noble?

Answer to comment 16: Modified.

Comment 17: Line 160: orexizing? oxidate?

Answer to comment 17: Please see answer to comment 10 and 11.

Comment 18: Line 173: peripheral organs to be oxidate [oxidized}

Answer to comment 18: Please see answer to comment 11.

Comment 19: Line 177 caloric restriction 

Answer to comment 19: Corrected, thanks.

Comment 20: Line 178: via mediated ?

Answer to comment 20: Please see answer to comment 12.

Comment 21: Line 178 inhibition of… 

Answer to comment 21: Corrected.

Comment 22: Line 205 – states regular menstrual cycles, but line 217 states by Rotterdam. Were women with regular menstrual cycles included or not?? Was it based on anovulation ie a progesterone level < … 

All baseline characteristics of the cohort should be presented in a baseline characteristics table eg AMH, cycle length, androgens etc

Answer to comment 22: In our study, the patients enrolled had a regular cycle but low levels of progesterone in the mid-luteal phase. While the length of the menstrual cycle was normal (within 25 to 35 days, patients had low progesterone values at the 21st day).

Comment 23: Line 218 why would family history of T2DM be an essential criteria for the cohort ?

Answer to comment 23: PCOS is disease whose etiology is due to a metabolic dysfunction. Family history of T2DM predispose to hyperinsulinemia/insulin resistance which, in turn, disturb ovarian function leading to hyperandrogenism (doi: 10.3390/biomedicines10081924).

Comment 24: Line 224 : incomplete

Answer to comment 24: All the patients selection criteria were listed.

Comment 25: Line 237 just to clarify the bloods were analysed immediately rather than being frozen. 

Answer to comment 25: The hormones were analyzed immediately, and the samples were not frozen.

Comment 26: Line 248 please clarify the duration of each phase. 

Answer to comment 26: Each phase had a duration of 4 weeks.

Comment 27: Line 263 please present the results of this study.

Answer to comment 27: done

Comment 28: Figure 1- Can you also present % with progesterone >15.9 nmol/L as per Leiva reference on threshold for ovulation. Please state how outliers were defined. 

Answer to comment 28: Done as requested (please see lines 302-304).

Comment 29: Why are Mean ±SD used in the text, but median plotted on box plots. What was the distribution of the data parametric ? 

Answer to comment 29: All data were normally distributed, therefore we used mean ± SD in the tables. Dot plots showing median, IQR, max and min values better represent the value distribution.

Comment 30: Line 301 oligomerrhea spelt wrong and did patients not have regular cycles in this study?

Answer to comment 30: Corrected. In our study, the patients enrolled had a regular cycle but low levels of progesterone in the mid-luteal phase

Comment 31: Line 309 did you intend to say 'exacerbated' by rather being a 'consequence' as such. Low SHBG is a consequence of PCOS and obesity / insulin resistance also, but in turn this increase free androgens. I see you are using a mendelian randomisation study to suggest causality, but there remains a bidirectional relationship which is well recognised.

Answer to comment 31: The relationship is bidirectional. We replaced “consequence” with “exacerbated”.

Comment 32: Line 310: which is currently, considered 

Answer to comment 32: Corrected. Thank you

Comment 33: Lin 362 how much weight was lost in this study? Please add to results table. 

Answer to comment 33: done

Comment 34: Line 392 and 394 Add refs

Answer to comment 34: These sentences have been deleted.

Reviewer 2 Report

CONGRATULATIONS

PCO syndrome is a complex, multifactorial disorder of still unknown etiology

„Does the ketogenic diet improve the quality of ovarian function in obese women?”

-the answer to this question is also very important as a voice in the discussion about PCO

Further studies are needed and a larger test group is needed, but it is a very interesting and innovative article

Author Response

Answers to Reviewer #2 comments

Manuscript ID: nutrients-1867796

Comment 1. CONGRATULATIONS. PCO syndrome is a complex, multifactorial disorder of still unknown etiology. „Does the ketogenic diet improve the quality of ovarian function in obese women?”. The answer to this question is also very important as a voice in the discussion about PCO. Further studies are needed and a larger test group is needed, but it is a very interesting and innovative article

Answer to comment 1. Dear Reviewer, we would like to thank you for the time you spent in reviewing our article, as well as for your comment. The limitation you highlighted has been added in the revised version of the manuscript (please see line 499). Best regards.

Round 2

Reviewer 1 Report

Please add the progesterone level that you used to state that patients were not ovulating, and that the patients had regular cycles. 

The units for the progesterone added is incorrect. 15.9 refers to nmol/L, can use ng/ml but would be a different number closer to 5. 

Author Response

Comment 1: Please add the progesterone level that you used to state that patients were not ovulating, and that the patients had regular cycles. 

Answer to comment 1: It is <10 ng/ml (please see Table 2)

Comment 2: The units for the progesterone added is incorrect. 15.9 refers to nmol/L, can use ng/ml but would be a different number closer to 5. 

Answer to comment 1. We confirm that ng/ml is correct.